# Characterization and Potential Action Mode Divergences of Homologous ACO1 Genes during the Organ Development and Ripening Process between Non-Climacteric Grape and Climacteric Peach

**DOI:** 10.3390/ijms25020789

**Published:** 2024-01-08

**Authors:** Linjia Luo, Pengcheng Zhao, Ziwen Su, Yuqing Huang, Yanping Zhang, Qian Mu, Xuxian Xuan, Ziyang Qu, Mucheng Yu, Ziyang Qi, Rana Badar Aziz, Peijie Gong, Zhenqiang Xie, Jinggui Fang, Chen Wang

**Affiliations:** College of Horticulture, Nanjing Agricultural University, Nanjing 210095, China; 2022104034@stu.njau.edu.cn (L.L.); zhaopengcheng@jsafc.edu.cn (P.Z.); 2018804207@njau.edu.cn (Z.S.); 2021204048@stu.njau.edu.cn (Y.H.); ypzhang@szai.edu.cn (Y.Z.); 2019104029@stu.njau.edu.cn (Q.M.); 2021204006@stu.njau.edu.cn (X.X.); qzyang@njau.edu.cn (Z.Q.); 2022804156@stu.njau.edu.cn (M.Y.); 2022804173@stu.njau.edu.cn (Z.Q.); azizranabadar@stu.njau.edu.cn (R.B.A.); gongpeijie@njau.edu.cn (P.G.); xiezhenqiang@jsafc.edu.cn (Z.X.); 2012088@njau.edu.cn (J.F.)

**Keywords:** grape, peach, ACO1, action modes, organ development and ripening, transient expression

## Abstract

Ethylene is one crucial phytohormone modulating plants’ organ development and ripening process, especially in fruits, but its action modes and discrepancies in non-climacteric grape and climacteric peach in these processes remain elusive. This work is focused on the action mode divergences of ethylene during the modulation of the organ development and ripening process in climacteric/non-climacteric plants. We characterized the key enzyme genes in the ethylene synthesis pathway, *VvACO1* and *PpACO1*, and uncovered that their sequence structures are highly conserved, although their promoters exhibit important divergences in the numbers and types of the cis-elements responsive to hormones, implying various responses to hormone signals. Subsequently, we found the two have similar expression modes in vegetative organ development but inverse patterns in reproductive ones, especially in fruits. Then, *VvACO1* and *PpACO1* were further validated in promoting fruit ripening functions through their transient over-expression/RNAi-expression in tomatoes, of which the former possesses a weaker role than the latter in the fruit ripening process. Our findings illuminated the divergence in the action patterns and function traits of the key *VvACO1/PpACO1* genes in the tissue development of climacteric/non-climacteric plants, and they have implications for further gaining insight into the interaction mechanism of ethylene signaling during the modulation of the organ development and ripening process in climacteric/non-climacteric plants.

## 1. Introduction

Ethylene is involved in modulating the organ development and ripening process, especially fruit ripening, which is a complex and dynamic process that involves a series of metabolic and physiological changes, including color, texture, flavor, and aroma [1,2]. According to the differences in ethylene release and the respiration intensity changes during the maturation and senescence process, fleshy fruits are traditionally classified into climacteric and non-climacteric [3]. The differences in ethylene synthesis and response between these two types of fruits imply that ethylene regulates the responsive genes through different mechanisms [4]. Climacteric fruits, such as tomato, apple, peach, kiwifruit, banana, and persimmon, have two different regulation systems for ethylene synthesis during the ripening process. System I functions during the early stage of fruit development and is responsible for producing basal levels of ethylene, while system II functions during fruit ripening and is responsible for producing high levels of ethylene [5]. Therefore, ethylene is the major factor that initiates and controls ripening in climacteric fruits. By contrast, non-climacteric fruits, such as citrus, strawberry, grape, loquat, and pineapple, only have a low level of ethylene production in system I during the maturation and senescence process, and their maturation does not strictly depend on ethylene [6]. However, recent studies exhibited that ethylene also plays an important role in some aspects of non-climacteric ripening [7,8]. For example, in non-climacteric grape, ethylene was required for its development and ripening, and exogenous ethylene remarkably enhanced the fruit color and anthocyanin content through up-regulation of the expression of genes related to anthocyanin biosynthesis [9,10]. Similarly, in strawberry, the increased expression of ethylene receptors was observed at the onset of ripening, and ethylene treatment also accelerated the sugar accumulation and fruit red-coloring and softening [11,12]. In citrus, ethylene enhanced the peel color break, and the respiration and ethylene production rates [13]. All these demonstrated ethylene is correlated closely with non-climacteric fruit development. In addition, research showed ethylene to be involved in the modulation of other tissues except for fruits in non-climacteric plants. For instance, it was reported that the key ethylene response factor for grape resistance to gray mold and clarified its regulatory mechanism in leaf resistance to gray mold [14], and ethylene significantly increased under low temperature stress and had a promoting effect on leaf cold resistance. Besides these, ethylene can induce pineapple flowering, presumably due to an increase in the endogenous ethylene content or increased sensitivity to endogenous ethylene [15,16]. In spite of these, little is known about the discrepancy of ethylene’s action modes in fruits and other tissues’ development and the ripening process between the climacteric and un-climacteric plants.

As is well known, 1-aminocyclopropane-1-carboxylic acid oxidase (ACO) is the last enzyme and key rate-limiting one in the ethylene synthesis process, which catalyzes the conversion of 1-aminocyclopropane-1-carboxylic acid (ACC) to ethylene [12]. Previous research reported that the ACO activity was constitutive, and ACS activity was widely known as the key determining enzyme in the control of ethylene production [17]. However, recently, the function of ACO in controlling ethylene biosynthesis has gradually become more conspicuous. In pre-climacteric tomato fruit, the rise in ACO activity preceded ACS activity in response to ethylene, indicating that ACO activity is related to the ethylene concentration [18]. Moreover, ACO is encoded by a small multi-gene family in various plant species, which has been characterized and identified in various climacteric and non-climacteric fruits, such as apple [19], strawberry [20], peach [21], kiwi [22], grape [23], and banana [24]. The expression patterns of ACO mRNA further supported the role of ACO in ethylene production during fruit ripening [17]. In tomatoes, there are five genes encoding ACO enzymes, of which three genes (*LeACO1*, *LeACO3*, *LeACO4*) are differentially expressed in fruits [25]. *LeACO1* and *LeACO4* accumulate during the immature period, and their expression level is significantly increased at the beginning of the mature period, while the expression of *LeACO1* and *LeACO4* remains unchanged during maturation, and the results suggest that the maturation-associated induction of *LeACO1* and *LeACO4* expression is dependent on ethylene [26]. In non-climacteric strawberry, Trainotti [27] cloned two *ACO* genes of *FaACO1* and *FaACO2*. These two genes were most expressed in flowers and less expressed in developing young fruits. Afterwards, from the bright green to white stage, *FaACO1* showed an increase in expression in red fruits and then continued to decrease until reaching its minimum value. In the case of the *FaACO1* gene, the minimum value appears in white fruits, followed by a sustained slight increase throughout the entire ripening process. Therefore, *FaACO1* might be one main factor in ethylene production in strawberry fruits. Except for fruits, *ACO* genes are also specifically expressed in other tissues. In apple, *MdACO1* is expressed in small amounts in young leaves, while *MdACO3* is mainly expressed in young and mature leaves, with almost no expression in young and mature fruit tissues [28]. In blueberries, *VcACO2* plays an important regulatory role in the development and maturation of the pistil, whose expression is 46.6 times higher in the pistil than in the stem, indicating there is a large amount of ethylene synthesis in the pistil during the reproductive growth of blueberries [29]. However, until now, the differences in ACO between climacteric and non-climacteric fruits are still inadequately understood, and much less is known about their action modes in tissue development and the ripening process.

Grapes and peaches are two economically important and widely cultivated fruit crops in the world [30,31]. Callhana [32] found that the expression levels of ACO genes in peaches varied among different tissue organs, mature fruits, and injured tissues. Previous studies showed that the expression of *PpACO1* and *PpACO2* in climacteric peach increased with fruit ripening, while the ACO activity and ethylene production also increased [33]. Chen [34] reported that peach fruit ripening is controlled by mediating the expression levels of genes such as *PpACO1*, which are involved in ethylene biosynthesis during ripening. However, in non-climacteric grape, the expression of *VvACO* and the level of ethylene only slightly increased at the veraison stage [23]. Cai [35] showed that *VvACO1* and *VvACO3* reach a high level in young grape stems, and *VvACO2* had the highest expression level in grape grains and cobs. During fruit development, the transcription peaks of *VvACO1* and *VvACO2* appeared at the fruit veraison stage, and the expression level of *VvACO1* was four times higher at veraison than at harvest time, while the expression level of *VvACO3* did not change significantly during the whole process [36]. All these suggest that *VvACO1* might be the key gene which causes the change in the ethylene content during grape tissue development and the ripening process [37], which could motivate us to further explore the action modes and differences of *ACO1* during this process in climacteric and non-climacteric plants.

This article is organized as follows. In Section 1, *VvACO1* and *PpACO1* are identified and characterized from grape and peach, including the gene and protein sequence, protein structure, phylogenetic relationship and promoter elements. In Section 2, the spatio-temporal expression patterns of *VvACO1* and *PpACO1* during various organs’ (especially reproductive ones) development processes are analyzed for recognizing their potential action mode variations via RNA-seq and qRT-PCR. In Section 3, the gene function divergence in promoting the fruit ripening process between *VvACO1* and *PpACO1* via transient over-expression/RNAi-expression in tomato. This work illuminates the divergence in the action modes and functions of *VvACO1* and *PpACO1* during organ development and the ripening process.

## 2. Results

### 2.1. Identification and Characterization of ACO1 Genes from Grape and Peach

To determine their roles in the tissues development and ripening process, VvACO1 and PpACO1 were isolated and identified from grape cv. ‘Fujiminori’ and peach cv. ‘Xiaobaifeng’ for thorough comparison and investigation. The open reading frame (ORF) of VvACO1 was 957 bp encoding 318 amino acids, and PpACO1 with 960 bp encoding 319 amino acids (Figure 1A,B). Totally, 53 amino acid changes were found through the amino acid sequence alignment between the VvACO1 and PpACO1 proteins. Among them, 32 amino acid variations are the same type, including 18 non-polar, 10 polar, 3 basic, and 1 acidic amino acid change; 21 amino acid variations are a different type, including 12 non-polar and 9 polar amino acid changes. To further recognize the functional conservation of ACO, we compared the sequences of the ACO proteins across eight different plant species. From the amino acid sequences analysis result, the ACO proteins presented high conservation across diverse plant species, with 87.73% identity in all eight sequences. Furthermore, these ACO proteins contained a highly conserved leucine zipper at the N-terminus, and thus the N-terminus was more conservative than the C-terminal, which was consistent with the previous report. From Figure 1C, it was observed that leucine appears once every seven amino acids in the primary structure of ACO (the location indicated by the red arrows). The Fe^2+^-binding mode and ascorbate-binding mode in the ACO proteins supported the fact that the enzyme activity of ACO requires Fe^2+^ and ascorbic acid as auxiliary factors. In addition, the sequence similarity of the ACOs from the climacteric group was high, the same as the non-climacteric group. It is speculated that the function of ACO belonging to the same respiratory climacteric type follows the same pattern associated with diversity within different types.

### 2.2. Phylogenetic and Conservative Analysis of VvACO1 and PpACO1 Proteins

The phylogenetic tree of homologous ACO1 across eight plant species was constructed via the Maximum Likelihood method (Figure 2A). In contrast with that by the Neighbor-Joining method (Appendix A), it was noted that the former method had a larger computational load and more accurate results than the latter method, so the former phylogenetic tree was further analyzed. The results showed that the eight ACO proteins from diverse plant species can be divided into two groups by the proximity of the evolutionary relationship (Figure 2A), and the ACO proteins from the plants with climacteric and non-climacteric fruits were grouped separately on two different branches without large gaps, and CcACO and MdACO were the closest relatives to VvACO1 and PpACO1, respectively. Combined with the domain analysis (Figure 2B), there are Pfam DIOX_N and Pfam 2OG_Fell_Oxy contained by all the proteins with almost identical length and position. The Pfam DIOX_N is a highly conserved N-terminal region of the proteins with 2-oxoglutarate/Fe(II)-dependent dioxygenase activity. And Pfam 2OG_Fell_Oxy is a domain enzyme with the Fe(2+) and 2-oxoglutarate (2OG)-dependent dioxygenase. In plants, the Fe(II) 2OG dioxygenase domain enzymes catalyze the formation of plant hormones, such as ethylene, gibberellins, anthocyanidins, and pigments. The purple square called the low complexity region, a region of low compositional complexity detected using the SEG program, is a universal existence in the protein structure. It is invisible because of the overlap with Pfam DIOX_N in PpACO1, PbACO1, and MdACO1.

The motif analysis was conducted to dig for more characteristics of the ACO proteins (Figure 2C,D). Totally, eight motifs were found in the ACO proteins. The result showed that each protein possessed all eight motifs except that FvACO1 and SlACO1 lack motif 8. Referring to the other seven motifs, they had almost no difference in the distribution of positions. Moreover, the sequences of motif 4, motif 5, and motif 6 were more conservative than those of the others, and the position of motif 5, half part of motif 2 and motif 6 just correspond to Pfam 2OG_Fell_Oxy. Concluding all the aspects of the comparison results, it can be elucidated that the ACO proteins are highly conservative in structure, implying high consistency in function.

The protein secondary structures of homologous ACO1 across the eight plant species were generated using the online software PRABI (https://npsa-pbil.ibcp.fr/cgi-bin/secpred_sopma.pl), of which the dominant structure in VvACO1 was alpha helix, accounting for 41.51%, followed by 35.22% random coil, and the extended strand and beta turn were relatively small, 16.98% and 6.29%, respectively. Similarly, the PpACO1 protein were mainly composed of alpha helix (41.07%) and random coil (33.86%), followed by extended strand (17.87%) and beta turn (7.21%) (Appendix A). Meanwhile, their secondary structures generated using the online software PSIPRED 4.0 (http://bioinf.cs.ucl.ac.uk/psipred/) also exhibited the high conservation in the dominate structures, like VvACO1 and PpACO1 with helix, 34.91%/36.99%; coil, 50%/49.22%; strand, 15.09%/13.79%, respectively (Appendix A), further supporting the results in Appendix A. Subsequently, the tertiary structures of the homologous ACO1 across the eight plant species were further predicted using the online software SWISS-MODEL (Figure 2E), which exhibited a parallel prediction. All these similarities of the homologous ACO1 across the eight plant species in the secondary and tertiary structures indicate that they possess the high conservation in the evolution of their structures.

### 2.3. The Promoter Analysis of VvACO1 and PpACO1

To recognize the potential functions of *VvACO1* and *PpACO1*, we analyzed the types and quantities of their motifs and the divergence motifs in their promoters. Based on the potential functions of the motifs in their promoters, the motifs could be classified into five types, which included light-related elements, hormone-related elements, stress-related elements, tissue-specific elements, and circadian elements (Figure 3). The elements of *VvACO1* were fewer than *PpACO1* significantly, and the reason may be that a third part of the *VvACO1* promoter sequence is an unsure ‘N’ sequence. In the prediction of two genes, the number of light-related elements was the highest, which might be attributed to the importance of photosynthesis as an essential component of the green plant. The number of hormone-related elements was the second highest. Interestingly, it was revealed that the promoters of both the *VvACO1* and *PpACO1* genes contained a certain number of cis-acting elements involved in ethylene and gibberellin responsiveness. Moreover, the promoter of *VvACO1* had an O_2_ site, which was responsive to zeatin (ZT), while the promoter of *PpACO1* had a TGA element, AuxRR core, CGTCA motif and ABRE, which were responsive to auxin (AUX), methyl jasmonate (MeJA), and abscisic acid (ABA). Among these tissue-specific elements and stress-related elements, both *VvACO1* and *PpACO1* contained a certain number of elements related to meristem expression (CAT-box or CCGTCC-box), endosperm expression (Skn-1_motif), anaerobic induction (ARE), and the fungal elicitor (Box-W1). In addition, the promoter of *VvACO1* contained an MBS element, the MYB binding site involved in drought-inducibility, disclosing its participation in the stress-response mechanism in MYB regulation, while the promoter of *PpACO1* contained circadian, forming the possibility that it may be related to the plant’s physiological cycles.

### 2.4. The Tissue-Specific Expression Modes of VvACO1 and PpACO1

Based on the transcriptome data of 54 tissues of grape, we analyzed the temporal and spatial expression patterns of the *VvACO1* gene. As shown in Figure 4, *VvACO1* was expressed in 54 tissues, organs, and development stages. In all the vegetative organs, the expression of *VvACO1* was higher in the mature leaf and tendril (the paralogous organ of flower), but lower in the root. In different tissues of flower organs, the expression of *VvACO1* in the petal was the highest, while the expression of *VvACO1* in the stamen was the lowest. During the development of the flower, *VvACO1* showed an overall increased expression pattern, the expression of *VvACO1* during young inflorescence was lowest. Similarly, the expressions of *VvACO1* in diverse tissues of the fruit organs were different, the expression in berry flesh was significantly higher than that in berry skin. In addition, the expression levels of *VvACO1* in berry pericarp, berry flesh, and berry skin were all highly expressed at the young berry post-fruit set, and they showed an overall decreased expression pattern during the development and ripening of the fruit. From the expression profiles, it was observed that *VvACO1* had a typical increased trend in its expression levels during the development and ripening process in the tendril and flower organs, but a decreased trend during the grape berry development and ripening process.

Given the expression characterization of *VvACO1* in the reproductive organs mentioned above, we further analyzed the temporal and spatial expression modes of *PpACO1* in peach flower buds, flowers and fruits by means of qRT-PCR. As shown in Figure 5, *PpACO1* was expressed differently in the flower and fruit organs at various stages, and it was expressed higher in fruit but lower in flower. However, throughout the developmental stages of the peach flower and fruit, *PpACO1* exhibited an overall rising expressional trend, especially during the fruit development and ripening process, appearing a drastically increasing trend and reaching the highest level at 90DAF (mature stage), indicating that *PpACO1* was actively involved in the later stages of peach flower and fruit ripening.

Further comparing the tissue-specific expression profiles of *VvACO1* and *PpACO1* examined above, we revealed that *ACO1* possessed the typical opposite expression patterns during fruits development between non-climacteric grape and climacteric peach, where *VvACO1* exhibited a sharp downward trend at berry pericarp and fresh tissues from grape post-fruit set to ripening, while *PpACO1* had a rapid upward one from young to ripening fruit, implying the significant divergence of ethylene-mediating tissues ripening specifically in climacteric/non-climacteric fruit development. Therefore, this provokes us to further investigate their function difference in fruits.

### 2.5. Transient Over-Expressed VvACO1 and PpACO1 in Tomato Fruits

To further validate the roles of *VvACO1* and *PpACO1* in fruits, we injected the *VvACO1* and *PpACO1* over-expression construct into tomato fruits at the late green fruit stage (about 15 days after fruit setting), using the empty pCAMBIA1302 vector as a control. As shown in Figure 6A, over-expression of *VvACO1* and *PpACO1* accelerated significantly the progress of fruit ripening. After 5 days of fruit injection, the *VvACO1*-OE and *PpACO1*-OE fruits started to become red, and after 10 days of fruit injection, the *VvACO1*-OE and *PpACO1*-OE fruits were completely mature, and the *PpACO1*-OE fruits had a redder color than the *VvACO1*-OE fruits, whereas the control fruit was just starting to turn red. Moreover, the PCR results showed that both the *VvACO1* and *PpACO1* genes were expressed in tomato (Figure 6B). In the *VvACO1*-OE and *PpACO1*-OE fruits, the expression of *SlACO1* was significantly up-regulated. In particular, the *PpACO1*-OE fruits resulted in a higher transcript level of *SlACO1* than the *VvACO1*-OE fruits (Figure 6B). Therefore, we concluded that *ACO1* played an important role in the regulation of fruit development and ripening, and the diverse *ACO1s* from the climacteric and non-climacteric fruits played positive roles to various extents during the fruit ripening processes.

### 2.6. Virus-Induced VvACO1 and PpACO1 Silencing in Tomato Fruits

The VIGS constructs that targeted *VvACO1* and *PpACO1* were generated and placed under the control of the cauliflower mosaic virus (CaMV) 35S promoter using the digestion of XbaI and BamHI. Transient silencing was performed by injecting the TRV1+*VvACO1*-TRV2 and TRV1+*PpACO1*-TRV2 constructs into tomato fruits at the middle of the green fruit stage (about 10 days after fruit setting), with an empty pCAMBIA1302 vector used as a control. After 25 days of fruit injection, significant differences in the phenotype were observed between the silencing fruits and control fruits, indicating the effect of *VvACO1*-silencing and *PpACO1*-silencing on the tomato fruits. As shown in Figure 7A, *VvACO1*-silencing and *PpACO1*-silencing inhibited fruit ripening, which was reflected by the pale color, whereas the control fruit matured normally, which showed as fully reddened. Furthermore, we conducted PCR detection with TRV2 primers to test whether the TRV vector can directly infect tomato fruit. As demonstrated in Figure 7B, the PCR product was present only in the TRV-infiltrated fruit and absent in the untreated fruit, indicating that recombinant TRV can efficiently disseminate and replicate in tomato fruit, and TRV-based vectors can be used to establish the VIGS system in tomato fruit. To confirm the *SlACO1* suppression at the molecular level, we performed qRT-PCR to detect the variation between the control and treated fruit (Figure 7B). Compared to control fruits, the expression of *SlACO1* was significantly down-regulated in the *VvACO1*-silencing and *PpACO1*-silencing fruits, suggesting that *VvACO1-* and *PpACO1*-silencing delayed maturation by inhibiting the expression of *SlACO1* in the tomato fruit.

## 3. Discussion

A fruit’s maturity is a fundamental determinant of its commercial quality [38], while plant hormones play a key role in the regulation of fruit ripening. Generally speaking, the ripening of climacteric fruits is mainly induced by the gaseous hormone ethylene [39], while the ripening of non-climacteric fruits is mainly regulated by ABA [40,41]. However, increasing evidence suggests that many fruit quality changes in non-climacteric fruits are also directly regulated by ethylene. Therefore, it is necessary to deeply understand the differences in the regulatory mechanisms of ethylene during the ripening process of climacteric and non-climacteric fruits.

### 3.1. Characterization and Expression Modes of VvACO1 and PpACO1 Genes

In this study, *PpACO1* from climacteric fruit peach and *VvACO1* from non-climacteric fruit grape were used as research materials to dig deeper into their roles in the ethylene synthesis pathway. First, *PpACO1* and *VvACO1* were cloned, sequenced, and compared with six ACO proteins in various other species. The most striking feature is that all the ACO proteins have a leucine zipper, a Fe^2+^-binding mode, and an ascorbate-binding mode. The Fe^2+^-binding mode and ascorbate-binding mode are completely conservative. There are Pfam DIOX_N located in the N-terminal with 2-oxoglutarate/Fe(II)-dependent dioxygenase activity and Pfam 2OG_Fell_Oxy with the Fe(2+) and 2-oxoglutarate (2OG)-dependent dioxygenase found as two conserved domains. Pfam 2OG_Fell_Oxy, a domain enzyme, typically catalyzes the oxidation of an organic substrate using a dioxygen molecule, mostly by using ferrous iron as the active site cofactor and 2OG as a co-substrate decarboxylated to succinate and CO_2_ [42,43]. In plants, Fe(II) 2OG dioxygenase domain enzymes catalyze the formation of plant hormones, such as ethylene, gibberellins, anthocyanidins, and pigments [44]. This is consistent with the substrate and reaction conditions of ethylene synthesis, so Pfam 2OG_Fell_Oxy should be the functional center of ACO proteins. The position of the low complexity region in climacteric fruit is closer to the N-terminal, and whether this will affect its function remains to be further studied.

*ACO* is a small multi-gene family, and different members exhibit spatiotemporal expression specificity [45]. In apple, *MdACO1* was mainly expressed in mature fruits, *MdACO2* was expressed in young fruits and leaves, and *MdACO3* was expressed in young and mature leaves [46]. In pear, although both *PpACO1* and *PpACO2* were expressed in the fruits, the product of *PpACO1* was mainly expressed in mature and senescent fruits, while *PpACO2* exhibited a high level of expression at the initial stage of fruit formation [47]. In tomato, *LeACO1* and *LeACO3* were mainly expressed in roots, *LeACO2* was predominantly expressed in flowers, and *LeACO4* was primarily expressed in mature fruits [48]. In this study, we analyzed the spatiotemporal expression pattern of the *VvACO1* gene in 54 tissues and developmental stages of grape based on transcriptome data. *VvACO1* was expressed in different organs at various stages of the grape and did not exhibit tissue-specific expression. Among all the vegetative organs, *VvACO1* exhibited higher expression in mature leaves and tendrils, but lower expression in roots. During flower development, *VvACO1* showed an overall increasing expression pattern, with the lowest expression in young inflorescences; in contrast, *VvACO1* exhibited a decreasing expression pattern during berry development, with high expression in young fruits. These results were consistent with previous studies [36,49]. Furthermore, we analyzed the spatiotemporal expression characteristics of *PpACO1* in peach using qRT-PCR and found that *PpACO1* was mainly expressed in the fruits, with lower expression in the flowers. With the development of the fruits, the expression of *PpACO1* gradually increased and reached the highest level at the ripening stage. In summary, there were significant differences in the high expression periods of *VvACO1* and *PpACO1* in fruits, which was consistent with the period of ethylene release in fruits. Correspondingly, non-climacteric fruits such as grapes and strawberries exhibited a peak of ethylene production in early fruit development [50,51], while climacteric fruits such as bananas and tomatoes exhibited a peak of ethylene production in the ripening stage [52].

### 3.2. Regulatory Roles of Plant Hormones in Climacteric and Non-Climacteric Plants

As the key unit of transcriptional regulation, cis-elements participate in the regulation of molecular networks in many biological processes, and the analysis of cis-regulatory elements in gene promoters is essential for understanding the regulatory roles of gene expression [53]. Through promoter action element analysis, we found that the promoter sequences of *VvACO1* and *PpACO1* contained a large number of hormone-related elements. It was worth noting that both the *VvACO1* and *PpACO1* promoter regions contained ethylene-responsive elements, ERELEE4 (ATTTCAAA) for *VvACO1* and LECPLEACS2 (TAAAATAT) for *PpACO1*. Additionally, *VvACO1* contained a GCC-box (core sequence AGCCGCC), which was often bound by ERF transcription factors to activate or inhibit ET/JA-linked defense pathways [54]. Moreover, the promoter of *VvACO1* had an O2 site and TATC box, which were responsive to zeatin (ZT) and gibberellin (GA); while the promoter of *PpACO1* had a TGA element, AuxRR core, GARE motif, CGTCA motif and ABRE, which were responsive to auxin (AUX), gibberellin (GA), methyl jasmonate (MeJA), and abscisic acid (ABA). These findings proved that the ripening of non-climacteric fruits is controlled by the synergistic action of multiple hormones rather than by a single hormone. The combined effects of auxin, GA, MeJA, ethylene, and ABA played crucial roles in regulating fruit ripening, forming complex feedback and cross-regulated networks between the plant hormone signaling pathways. For example, ABA induced ethylene production by regulating the expression of ethylene synthesis-related genes such as *ACS* and *ACO* [55]. In apple, MeJA treatment enhanced the expression of *MdACO1* and *MdACO2* during the storage period [56]. In peach, the application of exogenous auxin induced high *PpACS1* and *PpACO1* expression levels, substantial ethylene production, and fruit softening [57].

Additionally, both climacteric and non-climacteric fruit ripening are influenced by environmental factors, including light, temperature, and water, which may affect ripening through independent signaling systems or changes in the hormone signaling pathways [1]. At present, it is widely believed that ethylene synthesis and the ethylene signal transduction pathways are involved in various stress responses in plants [58]. The promoter of *VvACO1* and *PpACO1* contained a certain number of stress-related elements, including anaerobic (ARE), drought (MBS), cold temperature (LTR), heat shock response element (HSE), defense and stress responsiveness (TC-rich repeats), and fungal elicitor (Box-W1). Similarly, previous studies have shown that the expression of the *ACO* genes was also induced by biotic and abiotic stresses, such as high salt, low temperature, drought, and pathogen infection [59,60]. These reports provided a reasonable explanation for the presence of numerous stress-related elements in the promoter regions of *VvACO1* and *PpACO1*, indicating that *VvACO1* and *PpACO1* play an important regulatory role in the stress response.

### 3.3. The Biological Functions of VvACO1 and PpACO1

As shown in Figure 3, Figure 4 and Figure 5, we found that both *VvACO1* and *PpACO1* contained ethylene response elements, with *VvACO1* exhibiting high expression in the young fruit stage and *PpACO1* exhibiting high expression in the fruit ripening stage, indicating that they participated in the regulation of ethylene at specific stages. Furthermore, we transferred them to tomato fruits for deeper transgenic validation. The transient expression of *VvACO1* and *PpACO1* up-regulated the expression of *SlACO1* in tomato and promoted fruit ripening, and the promotion effect of *PpACO1* was stronger than that of *VvACO1*. By contrast, the silence of *VvACO1* and *PpACO1* repressed the expression of *SlACO1* in tomato and delayed the ripening of the fruits, and their inhibition effects were similar. These results indicated that *ACO1* played a positive role in the ripening process, and *PpACO1* from climacteric peach fruits had a more significant impact on the ripening process than *VvACO1* from non-climacteric grape fruits. In view of this phenomenon, two distinct ethylene biosynthesis systems have been described [5]. System I is widely present in climacteric and non-climacteric fruits, responsible for producing basal levels of ethylene, whereas system II is specifically present in climacteric fruits, automatically stimulating high levels of ethylene production.

In summary, the major differences related to ethylene between climacteric and non-climacteric fruit are the presence or absence of autocatalytic ethylene production. Since the amount of ethylene produced during the ripening and aging process of non-climacteric fruits is much lower than that of climacteric fruits, ethylene is believed to play a very limited role in the ripening process of non-climacteric fruits. However, more and more studies have shown that some molecular regulatory pathways involved in the ripening and aging of non-climacteric fruits and climacteric fruits are very similar, and many quality changes in non-climacteric fruits are directly regulated by ethylene [61]. Therefore, the similarities and differences in the regulation of ethylene in non-climacteric and climacteric fruits are still worth considering and exploring.

## 4. Materials and Methods

### 4.1. Plant Materials

The experiment was conducted at the Jiangsu Academy of Agricultural Sciences, Nanjing, China. The grape cultivar ‘Fujiminori’ and peach cultivar ‘Xiaobaifeng’ were selected as plant materials. Based on the growth and development curve for peach, flower bud samples were collected at four stages: young flower bud (21 days before flowering, 21DBF), well-developed flower bud (14 days before flowering, 14DBF), early flowering (7 days before flowering, 7DBF), and flowering (0 days before flowering, 0DBF). Fruit samples were collected at four stages: the first exponential growth (20 days post anthesis, 20DPA), pit hardening (50 days post anthesis, 50DPA), the second exponential growth (70 days post anthesis, 70DPA), and mature stage (90 days post anthesis, 90DPA). All the samples were frozen immediately in liquid nitrogen and stored at −80 °C for further analysis.

In addition, the tomato cultivar ‘Lichun’ was used for the agroinfiltration experiments. Tomato seeds were purchased from the market and surface sterilized via soaking in hot water at 50 °C for 10 min. After immersion, tomato seeds of a similar size were placed on moist filter paper-covered Petri dishes and pregerminated for 3 to 5 d in the dark. After 3 weeks of germination, the seedlings were transplanted to culture dishes containing a soilless potting mixture and moved to the greenhouse. The greenhouse conditions were maintained at 20–25 °C and 70% humidity. And the tomato plants were cultivated under a 14 h light and 10 h dark regime.

### 4.2. RNA Isolation and cDNA Synthesis

The total RNAs were extracted from the fruit samples using the RNAprep Pure Plant Kit (TianGen, Beijing, China) according to the manufacturer’s instructions. The concentration of the total RNAs was measured with a NanoDrop One Spectrophotometer (Thermo Scientific, Waltham, MA, USA), and the integrity of the total RNAs was evaluated via 1.5% agarose gel electrophoresis. Additionally, the cDNA was synthesized from the total RNAs using the Hifair^®^ II 1st Strand cDNA Synthesis Kit (Yeasen, Shanghai, China) and stored at −40 °C according to the manufacturer’s instructions.

### 4.3. PCR Amplification of the VvACO1 and PpACO1 Open Reading Frame

PCR amplification was performed with cDNA as the template and the corresponding primers. The reaction system had a total volume of 25 μL, including 1.0 μL of cDNA sample, 12.5 μL of 2 × Hieff™ PCR Master Mix, 1.0 μL of the two gene-specific primers, and 9.5 μL of ddH_2_O. The corresponding reaction program consisted of a 30 s denaturation cycle at 94 °C, a 30 s annealing cycle at 58 °C, 2 min extension cycles at 72 °C, and a final 10 min extension cycle at 72 °C. The PCR products were verified through 1.5% agarose gel electrophoresis and recovered using the FastPure Gel DNA Extraction Mini Kit (Vazyme Biotech, Nanjing Co., Ltd., Nanjing, China) according to the instructions. 

### 4.4. The Bioinformatics Analysis of VvACO1 and PpACO1

The protein sequences of VvACO1 and PpACO1 were obtained from the NCBI (https://www.ncbi.nlm.nih.gov/guide/) (accessed on 10 November 2021) and Phytozome 12 (https://phytozome.jgi.doe.gov/pz/portal.html#) (accessed on 1 September 2021). Furthermore, the homologous protein sequences of VvACO1 and PpACO1 from the other 6 non-climacteric and climacteric species, including FvACO1 (Fragaria vesca, XP_004293334.1), PaACO1 (Prunus avium, XP_021823230.1), CcACO (Citrus clementina, XP_006449820.1), SlACO1 (Solanum lycopersicum, NP_001234024.2), MdACO (Malus domestica, XP_008348143.1) and PbACO (Pyrus × bretschneideri, XP_009365121.1), were obtained using the Blast function of the NCBI online website (https://blast.ncbi.nlm.nih.gov/Blast.cgi) (accessed on 15 September 2021). Sequence analyses of VvACO1 and PpACO1 were conducted using DNANMAN 8.0 software (accessed on 17 September 2021), including ORF sequence identification and multiple sequence alignment. A phylogenetic tree was constructed using the Maximum Likelihood method with MEGA 5.0 software (accessed on 18 September 2021), with a validation parameter of Bootstrap = 1000. The protein conservative domains were analyzed using the SMART online program (http://smart.embl-heidelberg.de/) (accessed on 1 October 2021). The conserved motifs of the proteins were identified using MEME 5.5.5 online software (http://meme-suite.org/tools/meme) (accessed on 10 October 2021). The protein secondary structures were predicted using PRABI (https://npsa-pbil.ibcp.fr/cgi-bin/secpred_sopma.pl) (accessed on 13 October 2021) and PSIPRED 4.0 (http://bioinf.cs.ucl.ac.uk/psipred/) (accessed on 14 October 2021), and the protein tertiary structures were predicted using SWISS-MODEL online software (https://swissmodel.expasy.org/interactive) (accessed on 17 October 2021). The cis-elements in the promoter regions were predicted using the PlantCARE online program (http://bioinformatics.psb.ugent.be/webtools/plantcare/html/) (accessed on 25 October 2021).

### 4.5. Expression Analysis of VvACO1 and PpACO1

To detect the expression profiles of the *VvACO1* gene in grape, transcriptome data containing the expression data of fifty-four tissues or organs during grape development were downloaded from the NCBI Gene Expression Omnibus (https://www.ncbi.nlm.nih.gov/geo/query/acc.cgi?acc=GSE36128) (accessed on 23 October 2017) [62], and the Reads Per Kilobase per Million mapped read (RPKM) values were used to estimate the gene expression levels.

The expression of *PpACO1* was detected using quantitative real-time PCR (qRT-PCR). The related primer sequences are shown in Appendix A. qRT-PCR was performed following the instructions for the Hieff^®^ qPCR SYBR^®^ Green Master Mix (Yeasen, Shanghai, China) with the Applied Biosystems 7500 Real-Time PCR System. The amplification system consisted of 20 μL:10 μL of Hieff^®^ qPCR SYBR^®^ Green Master Mix, 1 μL of cDNA, 0.8 μL of upstream and downstream primer, and 7.4 μL of ddH_2_O. The qRT-PCR cycle parameters were set to 95 °C for 5 min, followed by 40 cycles of heating at 95 °C for 15 s and annealing at 60 °C for 35 s. All the gene expression tests were performed with three biological replicates and three technical replicates. The relative expression levels were calculated with the formula 2^−ΔΔCT^ = normalized expression ratio.

### 4.6. Vector Construction of VvACO1 and PpACO1

The fusion expression vectors were constructed using the Clone Express^®^ Ⅱ One Step Cloning Kit (Vazyme, Nanjing, China). Firstly, the amplified target fragments containing NcoI and SpeI were connected to a pCAMBIA-1302 vector. Secondly, the amplified target fragments containing XbaI and BamHI were connected to a pTRV2 vector. Finally, all the plasmid vectors were transferred to the Agrobacterium strain GV3101, and the positive clones were identified through colony PCR. The related primer sequences are shown in Appendix A.

### 4.7. Transient Injection of the Tomato Fruits

Tomato fruit infiltration was performed as described by Ratcliff with some modifications. Agrobacterium strains containing pTRV1 or pTRV2 were used for the VIGS experiment, while Agrobacterium strains containing pCAMBIA-1302 were used for the transient over-expression experiment. The Agrobacterium strains were cultured in LB liquid medium containing 10 mM MES and 20 μM AS with kanamycin (50 μg/mL) and rifampin (50 μg/mL). After 24 h, the Agrobacterium cells were harvested and resuspended in the infiltration buffer (10 mM MgCl2, 10 mM MES, pH 5.6, and 150 μM AS) to a final OD600 of 1.0. The suspensions were shaken for 4–6 h at room temperature before infiltration. Each Agrobacterium strain containing pTRV1 and pTRV2 was mixed in a 1:1 ratio and infiltrated into the tomato fruit with a 1 mL needle-less syringe. Six uniformly sized fruit were used in each infiltration experiment, which was repeated three times. The infiltrated fruits were maintained for over 10 days in a plastic tray at 18 °C. Tomato fruits infiltrated with TRV or pCAMBIA-1302 alone were used as the control.

## 5. Conclusions

In this work, we characterized the sequences of *VvACO1* and *PpACO1* from climacteric/non-climacteric plants and found that *VvACO1* and *PpACO1* possess high similarity in their sequences and protein structures, although their promoters have important divergences in the numbers and types of cis-elements responsive to hormones. Moreover, these two genes have similar expression modes in vegetative organs development, but the inverse patterns in reproductive ones, especially in fruits. Furthermore, both *VvACO1* and *PpACO1* can promote fruit ripening functions through their transient over-expression/RNAi-expression in tomatoes, of which the former possesses the weaker role than the latter in the fruit ripening process. Our findings are helpful to further investigate the mechanisms and their discrepancy in the ethylene signaling-mediated organ development and ripening process of climacteric/non-climacteric plants.

## Figures and Tables

**Figure 1 ijms-25-00789-f001:**
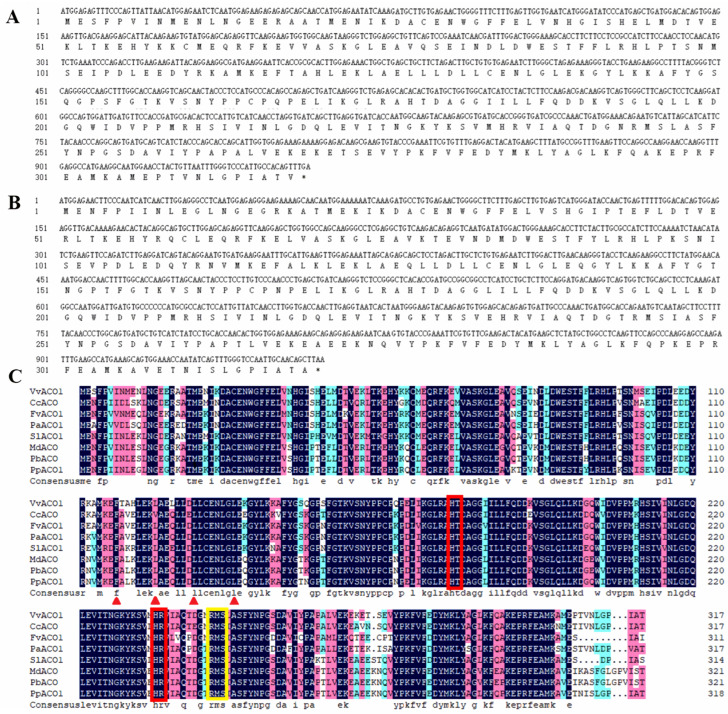
Sequence analysis of VvACO1 and PpACO1. (**A**) The ORF sequence of VvACO1. (**B**) The ORF sequence of PpACO1. The start codon and the stop codon were underlined. (**C**) Amino acid sequence analysis of 8 ACO proteins. Dark blue represents completely identical base sequences, pink represents partially identical base sequences, followed by sky blue, and white represents completely different base sequences. The red rectangles indicate the Fe^2+^-binding mode, and the orange rectangle indicates the ascorbate-binding mode.

**Figure 2 ijms-25-00789-f002:**
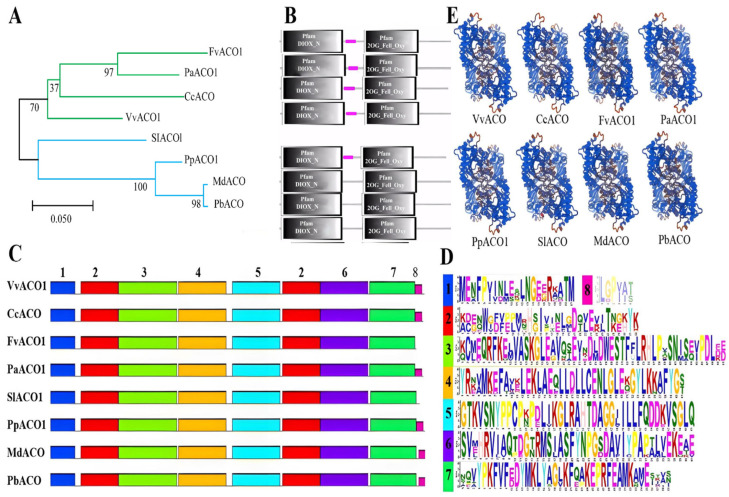
Phylogenetic and motif analysis of homologous ACO1 across 8 plant species. (**A**) Phylogenetic analysis. Blue and green represent different branches, black represents the main branch. (**B**) Domain analysis. (**C**) Motif analysis. The number of the motifs from left to right refers the position. (**D**) The sequences of the motifs, arranged according to the length of the motif sequences. (**E**) Protein tertiary structure analysis.

**Figure 3 ijms-25-00789-f003:**
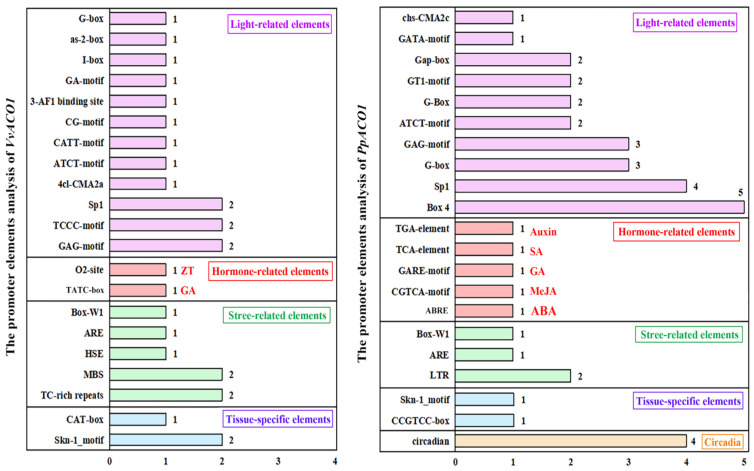
The promoter elements analysis of *VvACO1* and *PpACO1*. All the motifs of the promoters were classified into five types, including light, hormone, stress response, tissue-specific and circadian elements. Different colors indicate various types of elements.

**Figure 4 ijms-25-00789-f004:**
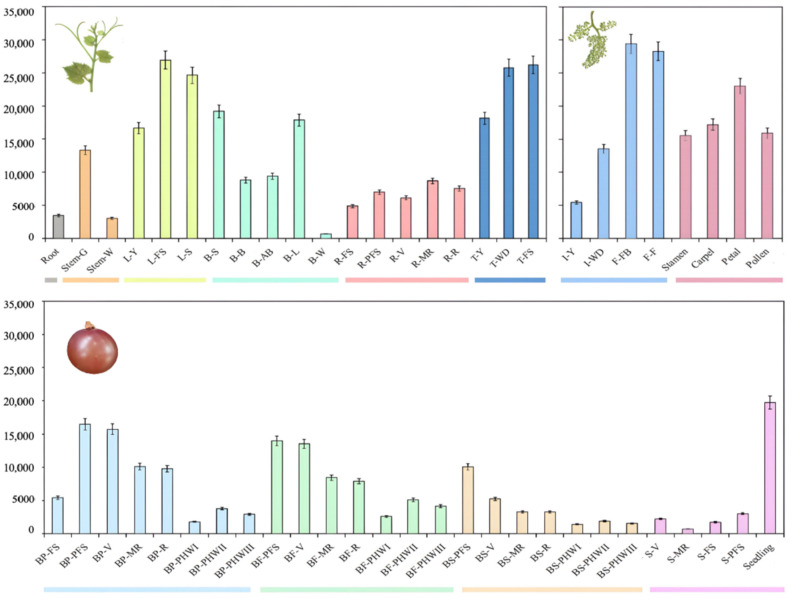
The expression of the *VvACO1* gene in different organs and tissues of grape. Notes: Root: root; Stem-G: green stem; Stem-W: woody stem; L-Y: young leaf; L-FS: mature leaf; L-S: senescence leaf; B-S: bud swell; B-B: bud burst; B-AB: bud after-burst; B-L: latent bud; B-W: winter bud; R-FS: rachis fruit set; R-PFS: rachis post fruit set; R-V: rachis veraison; R-MR: rachis mid-ripening; R-R: rachis ripening; T-Y: young tendril; T-WD: well-developed tendril; T-FS: mature tendril; I-Y: young inflorescence; I-WD: well-developed inflorescence; F-FB: flowering begins; F-F: flowering; Stamen: stamen; Carpel: carpel; Petal: petal; Pollen: pollen; BP-FS: berry pericarp fruit set; B-PFS: berry pericarp post-fruit set; BP-V: berry pericarp veraison; BP-MR: berry pericarp mid-ripening; BP-R: berry pericarp ripening; BP-P I: berry pericarp post-harvest withering I; BP-P II: berry pericarp post-harvest withering II; BP-P III: berry pericarp post-harvest withering III; BF-PFS: berry flesh post-fruit set; BF-V: berry flesh veraison; BF-MR: berry flesh mid-ripening; BF-R: berry flesh ripening; BF-P I: berry flesh post-harvest withering I; BF-P II: berry flesh post-harvest withering II; BF-P III: berry flesh post-harvest withering III; BS-PFS: berry skin post-fruit set; BS-V: berry skin veraison; BS-MR: berry skin mid-ripening; BS-R: berry skin ripening; BS-PI: berry skin post-harvest withering I; BS-P II: berry skin post-harvest withering II;BS-P III: berry skin post-harvest withering III; S-V: seed veraison; S-MR: seed mid-ripening; S-FS: seed fruit set; S-PFS: seed post-fruit set; Seeding: seeding.

**Figure 5 ijms-25-00789-f005:**
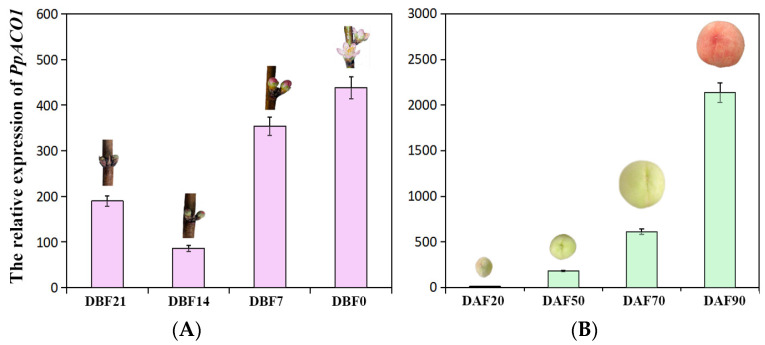
The expression of the *PpACO1* gene during the flower and fruit development of peach. (**A**) The expression of *PpACO1* at different time points (21, 14, 7, 0 days before flowering (21DBF, 14DBF, 7DBF, 0DBF)) of flower development. (**B**) The expression profiles of *PpACO1* at different time points (20, 50, 70, 90 days after flowering (20DAF, 50DAF, 70DAF, 90DAF)) of fruit development.

**Figure 6 ijms-25-00789-f006:**
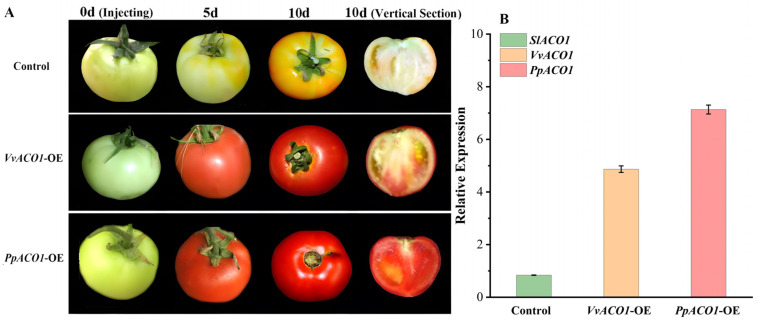
Transient over-expression of *VvACO1* and *PpACO1* in tomato fruit ripening. (**A**) Transient over-expression of *VvACO1* and *PpACO1* in tomato fruits. (**B**) The expression levels of *SlACO1/VvACO1/PpACO1* in the control and transgenic tomatoes.

**Figure 7 ijms-25-00789-f007:**
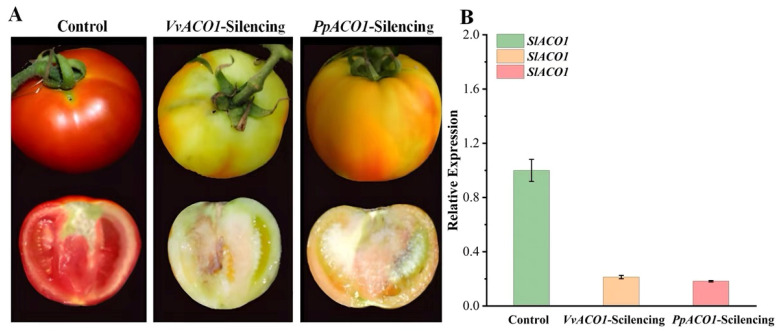
Transient silence of *VvACO1* and *PpACO1* in tomato fruits. (**A**) Color changes in tomato fruits infiltrated with Agrobacterium containing the empty vector TRV, TRV-*VvACO1*, and TRV-*PpACO1.* (**B**) The expression of *SlACO1* in the control and transgenic tomatoes.

## Data Availability

Data are available on request from the authors.

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
