# Peer review of "Characterization and Potential Action Mode Divergences of Homologous ACO1 Genes during the Organ Development and Ripening Process between Non-Climacteric Grape and Climacteric Peach"

_ijms, 2024, doi:10.3390/ijms25020789_

Round 1

Reviewer 1 Report

Comments and Suggestions for Authors

Review of the article "Characterization and potential action mode divergence of ACO1 between grape and peach during tissue development and ripening process"

In my opinion, the article is interesting and potentially valuable.

My points to improve the article:

1. The Abstract is long, but it does not emphasize novelties and the aim of the work.

   Please highlight the novelty in the Abstract and clearly define the aim of the work, i.e. "The aim of this article is ...".

2. The sentence "Our findings have important implications in the action mechanism of the ethylene during non-climacteric grape and climacteric peach tissue development and ripening processes." should be shifted from the Introduction to the Conclusions.

3. At the end of the Introduction, clear information about the organization of this article should be added, i.e.

   "This article is organized as follows. In section 2, ... In section 3, ..."

4. Qualities of all figures are bad and for this reason it is almost impossible to evaluate the parts of the results. The quality of all figures needs to be improved.

5. Section 2.2. Phylogenetic and Conservative Analysis of VvACO1 and PpACO1Proteins.

   a) To generate the phylogenetic tree the Authors used: "Neighbor-Joining (NJ) method by MEGA software with a validation parameter of Bootstrap=1000.", but reliabilities (calculated using the bootstrap method) of the tree nodes are unknown, i.e. reliabilities are not presented in Figure 2.

   b) The Authors should add the information about the method used to correct evolutionary distances and method used to align sequences.

   c) The tree generated using the Maximum Likelihood method (that is also in the used MEGA program) should be added and results should be compared with the results obtained using the Neighbor-Joining method.

6. A link to the PRABI program should be added.

   The protein secondary structures were predicted using the online software PRABI, but I do not see the modeled secondary structures in the article.

   Moreover, please use the PSIPRED program (http://bioinf.cs.ucl.ac.uk/psipred/) to obtain the secondary structures and compaire the results.

7. The SWISS-MODEL program was used to predict the tertiary structures.

   The specificity of SWISS-MODEL is that the modeled tertiary structures, in these cases presented in the article, will be the same or almost the same. This does not mean that these structures are the same (or almost the same), this is only due to the algorithm implemented in SWISS-MODEL.

   After modeling using SWISS-MODEL the Authors concluded "it was noted that their tertiary structures exhibit a parallel prediction, which may reflect a high similarity in function." - for the reason I wrote about, this conclusion can be untrue. 

8. There is (lines 433-435):

   "Based on previous studies (Figure 3, Figure 4, and Figure 5), we found that both VvACO1 and PpACO1 contained ethylene response elements, with VvACO1 exhibiting high expression in the young fruit stage and PpACO1 exhibiting high expression in the fruit ripening stage, indicating that they participated in the regulation of ethylene at specific stages."

   a) As I good understand, that means that in figures: Figure 3, Figure 4 and Figure 5, the results obtained in previous studies have been presented. Have these three figures also been presented in the previous published publications?

   b) References to published articles (i.e. the previous studies) should be added after this sentence.

9. There is lack of a Conclusions section in the article.

10. Linguistic mistakes (including typographical) require correction.

Comments on the Quality of English Language

Linguistic mistakes (including typographical) require correction.

Author Response

We would like to thank you for reviewing our work and giving us the detailed list of comments and suggestions. A detailed point-to-point response to you is given,Please see the attachment.

Reviewer 2 Report

Comments and Suggestions for Authors

The authors investigate the influence of phytohormones on plant development and ripening process. From that point of view, it is an interesting research that might be worth of publishing. However, the authors need to address some serious points before any final decision can be made: 

1) The manuscript has been written quite carelessly. A number of sentences are way too long and difficult to follow. For instance: 

 "Tissue-specific expression analysis showed VvACO1 had the much high expressions in mature tissues of leaves, tendrils, flower, but very low expressions in the corresponding young tissues (inflorescences are young flowers), interestingly, in various berry tissues, appearing the reverse modes that VvACO1 was higher at its expressions in young berries than mature ones, and exhibiting one the trend of decline from the berries post-fruit set to ones post-harvest; in contrast with the climacteric peach, further comparing to PpACO1 expressions in reproductive organs of flowers and fruits, PpACO1 showed a gradually in- creasing trend during the development of flower and fruit, of which the former is similar to those from inflorescences to flowering in grapes, while the laô€„´er showed the opposite trend to those during grape berries."

"Climacteric fruits, such as tomato, apple, peach, kiwifruit, banana, and persimmon, have two different regulation systems for ethylene synthesis during the ripening process: System I ethylene production, also known as the autoinhibitory loop, functions during the early stage of fruit development and is responsible for producing basal levels of ethylene; while system II ethylene production, also known as the auto-stimulatory loop, functions during fruit ripening, and is responsible for producing high levels of ethylene[29]."

Many empty spaces are missing in the manuscrip. For instance, when references are used, there should be an empty space between a word and [x]. Similarly, after dots, commas, between the words ("economicallyimportant and widelycultivated"), etc. 

Some abbreviations are written using normal font and then again using italic font. This must be unified. 

2) The abstract is way too long. The authors should make it more concise, mentioning the major topic of research, methods and achieved results. 

3) The second section after Introduction is Results. This is totally atypical. I would expect first to explain the methods and theoretical background, and to explain the research conducted before the authors come to conclusions. 

4) There are no conclusions. The paper needs to end with a section entitled Conclusions, which is not the same as Discussion. 

5) Most of the figures are of low quality and not readable, which is unacceptable.  

Comments on the Quality of English Language

Too long sentences, difficult to follow, many empty spaces missing, etc. 

Author Response

We would like to thank you for reviewing our work and giving us the detailed list of comments and suggestions. A detailed point-to-point response to you  is given,Please see the attachment, thank you.

Round 2

Reviewer 1 Report

Comments and Suggestions for Authors

The Authors have correctly addressed all my concerns and comments. The corrected article is better and, in my opinion, suitable for publication in the International Journal of Molecular Sciences.

Author Response

We really appreciated your positive evaluation and suggestion on acceptance of our manuscript.

Reviewer 2 Report

Comments and Suggestions for Authors

The authors did some effort to improve the paper, but it is still not acceptable. The major comments are: 

1) The quality of figures is still not at the required level. In some figures there is a text which I cannot read. The resolution of the figures and the font size must be selected so that the text is readable. 

2) Section Results as the 2nd section (after Introduction) and section Materials and Methods as the 4th section (just before Conclusions)? This is definitely not acceptable, as it makes no sense at all. The authors are strongly suggested to check other papers published in this and other journals in order to understand how a logical structure of a paper looks like. You cannot present results before you have presented materials and methods that you used to obtain those results. There has to be a logical order. 

3) Conclusions are unacceptable - in only 2 sentences, the authors simply repeat what has been done (1 sentence) and say what their findings are good for (1 sentence). The major part of conclusions must be what the general findings are, why those are helpful and their applicability. Conclusions must also include limitations of the work conducted and some clear directions for the future work in the field. Again, the authors are strongly suggested to check published papers and pay attention to the typical form of conclusions and information provided in this section. 

Author Response

We would like to thank you for reviewing our work and giving us the detailed list of comments and suggestions. A detailed point-to-point response to your comments is given, please see the attachment, thank you very much.

Round 3

Reviewer 2 Report

Comments and Suggestions for Authors

The paper can be published in its present form. 

Author Response

(The authors gave the same response as above.)
